# A crowdsourced analysis to identify ab initio molecular signatures predictive of susceptibility to viral infection

Slim Fourati [1], Aarthi Talla[1], Mehrad Mahmoudian[2,3], Joshua G. Burkhart [4,5], Riku Klén [2], Ricardo Henao [6,7], Thomas Yu [8], Zafer Aydın [9], Ka Yee Yeung [10], Mehmet Eren Ahsen [11], Reem Almugbel[10], Samad Jahandideh[12], Xiao Liang[10], Torbjörn E.M. Nordling [13], Motoki Shiga [14], Ana Stanescu [11,15], Robert Vogel[11,16], The Respiratory Viral DREAM Challenge Consortium[#], Gaurav Pandey [11], Christopher Chiu [17], Micah T. McClain[6,18,19], Christopher W. Woods[6,18,19], Geoffrey S. Ginsburg [6,19], Laura L. Elo[2], Ephraim L. Tsalik [6,19,20], Lara M. Mangravite [8] & Solveig K. Sieberts [8]

The response to respiratory viruses varies substantially between individuals, and there are currently no known molecular predictors from the early stages of infection. Here we conduct a community-based analysis to determine whether pre- or early post-exposure molecular factors could predict physiologic responses to viral exposure. Using peripheral blood gene expression profiles collected from healthy subjects prior to exposure to one of four respiratory viruses (H1N1, H3N2, Rhinovirus, and RSV), as well as up to 24 h following exposure, we find that it is possible to construct models predictive of symptomatic response using profiles even prior to viral exposure. Analysis of predictive gene features reveal little overlap among models; however, in aggregate, these genes are enriched for common pathways. Heme metabolism, the most significantly enriched pathway, is associated with a higher risk of developing symptoms following viral exposure. This study demonstrates that pre-exposure molecular predictors can be identified and improves our understanding of the mechanisms of response to respiratory viruses.

[1] Department of Pathology, School of Medicine, Case Western Reserve University, Cleveland, OH 44106, USA. [2] Turku Centre for Biotechnology, University of Turku and Åbo Akademi University, FI-20520 Turku, Finland. [3] Department of Future Technologies, University of Turku, FI-20014 Turku, Finland. [4] Department of Medical Informatics and Clinical Epidemiology, School of Medicine, Oregon Health & Science University, Portland, OR 97239, USA. [5] Laboratory of Evolutionary Genetics, Institute of Ecology and Evolution, University of Oregon, Eugene, OR 97403, USA. [6] Duke Center for Applied Genomics and Precision Medicine, Duke University School of Medicine, Durham, NC 27710, USA. [7] Department of Electrical and Computer Engineering, Duke University, Durham, NC 27708, USA. [8] Sage Bionetworks, Seattle, WA 98121, USA. [9] Department of Computer Engineering, Abdullah Gul University, Kayseri 38080, Turkey. [10] School of Engineering and Technology, University of Washington Tacoma, Tacoma, WA 98402, USA. [11] Department of Genetics and Genomic Sciences and Icahn Institute for Genomics and Multiscale Biology, Icahn School of Medicine at Mount Sinai, New York, NY 10029, USA. [12] Origent Data Sciences, Inc., Vienna, VA 22182, USA. [13] Department of Mechanical Engineering, National Cheng Kung University, Tainan 70101, Taiwan. [14] Department of Electrical, Electronic and Computer Engineering, Faculty of Engineering, Gifu University, Gifu 501-1193, Japan. [15] Department of Computer Science, University of West Georgia, Carrolton, GA 30116, USA. [16] IBM T.J. Watson Research Center, Yorktown Heights, NY 10598, USA. [17] Section of Infectious Diseases and Immunity, Imperial College London, London W12 0NN, UK. [18] Medical Service, Durham VA Health Care System, Durham, NC 27705, USA. [19] Department of Medicine, Duke University School of Medicine, Durham, NC 27710, USA. [20] Emergency Medicine Service, Durham VA Health Care System, Durham, NC 27705, USA. These authors contributed equally: Slim Fourati, Aarthi Talla, Mehrad Mahmoudian, Joshua G. Burkhart, Riku Klén. [#]A full list of consortium members appears at the end of this paper. Correspondence and requests for materials should be addressed to L.M.M. (email: lara.mangravite@sagebase.org) or to S.K.S. (email: solly.sieberts@sagebase.org)

Acute respiratory viral infections are among the most common reasons for outpatient clinical encounters[1]. Symptoms of viral infection may range from mild (e.g. sneezing, runny nose) to life-threatening (dehydration, seizures, death), though many individuals exposed to respiratory viruses remain entirely asymptomatic[2]. Variability in individuals' responses to exposure has been observed both in natural infections[3] and controlled human viral exposure studies. Specifically, some individuals remained asymptomatic despite exposure to respiratory viruses, including human rhinovirus (HRV)[4–6], respiratory syncytial virus (RSV)[4–6], influenza H3N2[4–9], and influenza H1N1[4,5,9]. Factors responsible for mediating response to respiratory viral exposure are poorly understood. These individual responses are likely influenced by multiple processes, including the host genetics[10], the basal state of the host upon exposure[11], and the dynamics of host immune response in the early hours immediately following exposure and throughout the infection[12]. Many of these processes occur in the peripheral blood through activation and recruitment of circulating immune cells[13]. However, it remains unknown whether host factors conferring resilience or susceptibility to symptomatic infectious disease can be detected in peripheral blood before infection, or whether they are only apparent in response to pathogen exposure.

In order to identify such gene expression markers of resilience and susceptibility to acute respiratory viral infection, we utilized gene expression data from seven human viral exposure experiments[6,7,9]. These exposure studies have shown that global gene expression patterns measured in peripheral blood around the time of symptom onset (as early as 36 h after viral exposure) are highly correlated with symptomatic manifestations of illness[6,9]. However, these later-stage observations do not necessarily reflect the spectrum of early timepoint immune processes that might predict eventual infection. Since transcriptomic signals are weak at these early timepoints, the detection of early predictors of viral response has not yet been possible in any individual study. By combining data collected across these seven studies and leveraging the community to implement state-of-the-art analytical algorithms, the Respiratory Viral DREAM Challenge (www.synapse.org/ViralChallenge) aims to develop early predictors of resilience or susceptibility to symptomatic manifestation based on expression profiles that are collected prior to and at early timepoints following viral exposure and to understand the biological mechanisms underlying those predictors.

## Results

**Human viral exposure experiments.** In order to determine whether viral susceptibility could be predicted prior to viral exposure, we collated seven human viral exposure experiments: one RSV, two influenza H1N1, two influenza H3N2, and two HRV studies, in which a combined total of 148 healthy volunteers were exposed to virus (Supplementary Data 1; Fig. 1a–c) or sham ($n = 7$)[6,7,9]. Subjects were excluded if pre-existing neutralizing antibodies were detected, except for the RSV study in which neutralizing antibodies were not an exclusion criteria. Each subject in the study was followed for up to 12 days after exposure and serially sampled for peripheral blood gene expression by Affymetrix Human U133A 2.0 GeneChips. Throughout the trial, subjects self-reported clinical symptom scores across 8−10 symptoms (Supplementary Figure 1). These data were used to stratify subjects as either symptomatic or asymptomatic and to quantify symptom severity. Additionally, nasopharyngeal swabs measured viral shedding; these data were used to stratify subjects as either shedders or nonshedders (Fig. 1d). Clinical symptoms were summarized based on a modified Jackson score[14] and viral shedding was determined to be present if two or more measurable titers or one elevated titer was observed within 24 h following viral exposure[15]. Viral shedding and clinical symptoms

were provided to the Respiratory Viral DREAM Challenge participating teams only for the training data set (Fig. 1b). An additional, but not previously available, human exposure experiment to the RSV virus ($n = 21$) was used as an independent test data set (Fig. 1b, c). The study design for this data set was similar to those of the seven original data sets.

**Data analysis challenge.** Using these data, an open data analysis challenge, the Respiratory Viral DREAM Challenge, was formulated. Teams were asked to predict viral shedding and clinical symptoms based on peripheral blood gene expression data from up to two timepoints: prior to viral exposure ($T_0$) or up to 24 h post viral exposure ($T_{24}$). Based on gene expression data from the two timepoints, teams were asked to predict at least one of three outcomes: presence of viral shedding (subchallenge 1 (SC1)), presence of symptoms, defined as a modified Jackson score $\geq 6$ (subchallenge 2 (SC2)), or symptom severity, defined as the logarithm of the modified Jackson score (subchallenge 3 (SC3)). Teams were asked to submit predictions based on gene expression and basic demographic (age and gender) data from both timepoints to enable cross-timepoint comparison. The seven collated data sets served as a training data set on which teams could build their predictive models. For a subset of subjects ($n = 23$), phenotypic data were withheld to serve as a leaderboard test set for evaluation with real-time feedback to teams (Fig. 1a).

Teams were asked to submit at least one leaderboard submission at each timepoint to be evaluated on the leaderboard test set. Performance metrics for these models were returned in real time, and teams could update their submissions accordingly up to a maximum of six combined submissions per subchallenge. At the end of this exercise, teams were asked to provide leave-one-out cross-validation-based predictions on the training set (LOOCVs) and predictor lists for each of their best models.

Each team's best models (one for $T_0$ and one for $T_{24}$) per subchallenge were ultimately assessed on the held-out human RSV exposure data set that had not been publicly available, previously (Fig. 1a). Predictions for the binary outcomes (shedding and symptoms) were assessed using Area Under the Precision-Recall (AUPR) and Receiver Operating Characteristic (AUROC) curves, and ranked using the mean rank of these two measures. The predictions for the continuous outcome (symptom severity) were assessed using Pearson's correlation ($r$) with the observed values. In each case, permutation-based $p$ values were used to identify submissions that performed significantly better than those expected at random. In total, 37 teams participated in some stage of the challenge (Supplementary Table 1).

**Challenge results.** For presence of symptoms (SC2), 27 models were assessed on the independent test data; 13 models were developed using $T_0$ predictors, and 14 models using $T_{24}$ predictors. Four of the $T_0$ models and three of the $T_{24}$ models achieved a nominal $p$ value of 0.05 for AUPR or AUROC, with the best scoring models at each timepoint achieving similar scores (AUPR($T_0$) = 0.958, AUROC($T_0$) = 0.863, AUPR($T_{24}$) = 0.953, AUROC($T_{24}$) = 0.863). Team Schrodinger's Cat was the only team that achieved nominal significance for all measures and timepoints. Despite the few teams achieving statistical significance, the models submitted were overall more predictive than expected at random (one-sided Kolmogorov–Smirnov test for enrichment $p$ values 0.008, 0.002, 0.021, and 0.05 for AUPR($T_0$), AUROC($T_0$), AUPR($T_{24}$), and AUROC($T_{24}$), respectively; Fig. 2a).

For symptom severity (SC3), 23 models were assessed on the independent test data; 11 models were developed using $T_0$ predictors and 12 models using $T_{24}$ predictors. Four of the $T_0$ models and two of the $T_{24}$ models achieved a nominal $p$ value of

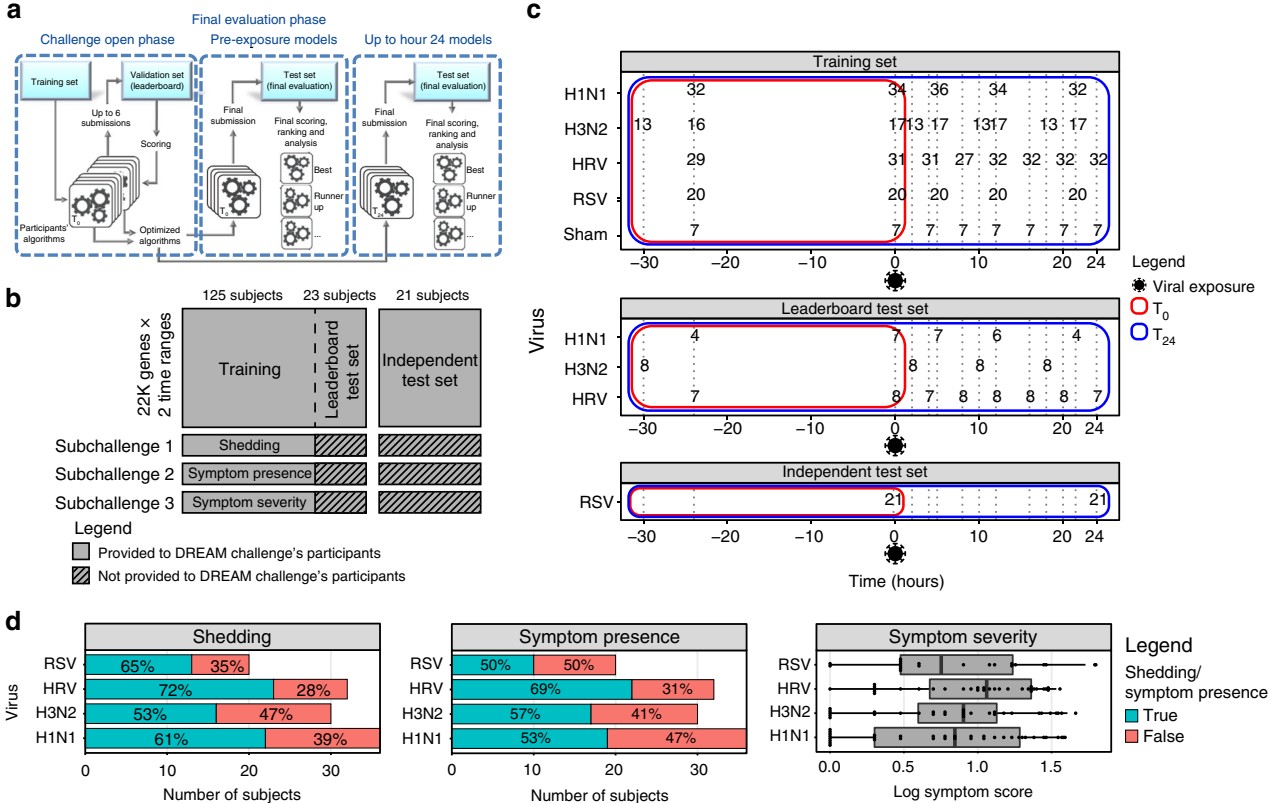

**Fig. 1** Respiratory Viral DREAM Challenge overview. **a** Schematic representation of the Respiratory Viral DREAM Challenge workflow. Participants used feedback from evaluation on the leaderboard test set to optimize their $T_0$ and $T_{24}$ models, and submitted a single model, per timepoint, for final evaluation on the Independent Test Set. **b** Schematic representing the data provided to participants. 125 subjects were provided as training data, 23 subjects were provided as a leaderboard test set, and 21 subjects from an independent data set were used for final evaluation. **c** Challenge data come from seven viral exposure trials with sham or one of four different respiratory viruses (H1N1, H3N2, Rhinovirus, and RSV). In each of these trials, healthy volunteers were followed for 7−9 days following controlled nasal exposure to one respiratory virus. Blood was collected and gene expression of peripheral blood was performed 1 day (24−30 h) prior to exposure, immediately prior to exposure and at regular intervals following exposure. Data were split into a training, leaderboard, and independent test set. Outcome data for the leaderboard and independent test set were not provided to the teams, but instead, teams were asked to predict them based on gene expression pre-exposure ($T_0$) or up to 24 h post-exposure ($T_{24}$). **d** Histograms and boxplot of the three outcomes by viruses. Symptom data and nasal lavage samples were collected from each subject on a repeated basis over the course of 7−9 days. Viral infection was quantified by measuring the release of viral particles from viral culture or by qRT-PCR (viral shedding). Symptomatic data were collected through self-report on a repeated basis. Symptoms were quantified using a modified Jackson score, which assessed the severity of eight upper respiratory symptoms (runny nose, cough, headache, malaise, myalgia, sneeze, sore throat, and stuffy nose). On the boxplot, the lower whisker, the lower hinge, the mid hinge, the upper hinge and the upper whisker correspond to −1.5× the interquartile (IQR) from the first quartile, the first quartile, the median, the third quartile and 1.5× IQR from the third quartile of the log symptom score, respectively

0.05 for correlation with the observed log-symptom score, and as above, the best performing models scored similarly at both timepoints ($r = 0.490$ and $0.495$ for $T_0$ and $T_{24}$, respectively). Teams cwruPatho and Schrodinger's Cat achieved significant scores at both timepoints. Consistent with SC2, we also saw that the models submitted were overall more predictive than expected at random (one-sided Kolmogorov−Smirnov test for enrichment $p$ values 0.005 and 0.035 for $T_0$ and $T_{24}$, respectively; Fig. 2b). For both SC2 and SC3, enrichment was more pronounced at $T_0$ compared to $T_{24}$. Correlation between final scores and leaderboard scores was higher at $T_0$, suggesting $T_{24}$ predictions may have been subject to a greater degree of overfitting.

For viral shedding (SC1), 30 models were assessed from 16 different teams; 15 models were developed using $T_0$ predictors and 15 models using $T_{24}$ predictors. No submissions were statistically better than expected by random. In aggregate, these submissions showed no enrichment (one-sided Kolmogorov−Smirnov test for enrichment $p$ values 0.94, 0.95, 0.82, and 0.95, for AUPR($T_0$), AUROC($T_0$), AUPR($T_{24}$), and AUROC($T_{24}$), respectively). In contrast, final scores were negatively correlated with leaderboard

scores ($r = -0.22$, $-0.19$, $-0.65$, and $-0.54$ for AUPR($T_0$), AUROC($T_0$), AUPR($T_{24}$), and AUROC($T_{24}$), respectively) suggesting strong overfitting to the training data or a lack of correspondence to viral shedding as assessed in the independent test data set, relative to the training data sets. The negative correlation was strongest at $T_{24}$ (Supplementary Figure 2). Accordingly, results based on this subchallenge were excluded from further analysis.

**Best performing approaches.** The two overall best performing teams were Schrodinger's Cat and cwruPatho. Team Schrodinger's Cat used the provided gene expression profiles before the viral exposure to predict shedding and log symptom scores (binary and continuous outcomes, respectively). For the $T_0$ models, arithmetic means over measurements prior to exposure were calculated, whereas for the $T_{24}$ models, only the latest measurements before viral exposure were used. Epsilon support vector regression (epsilon-SVR)[16] with a radial kernel and tenfold cross-validation were used to develop the predictive models. Their work demonstrated

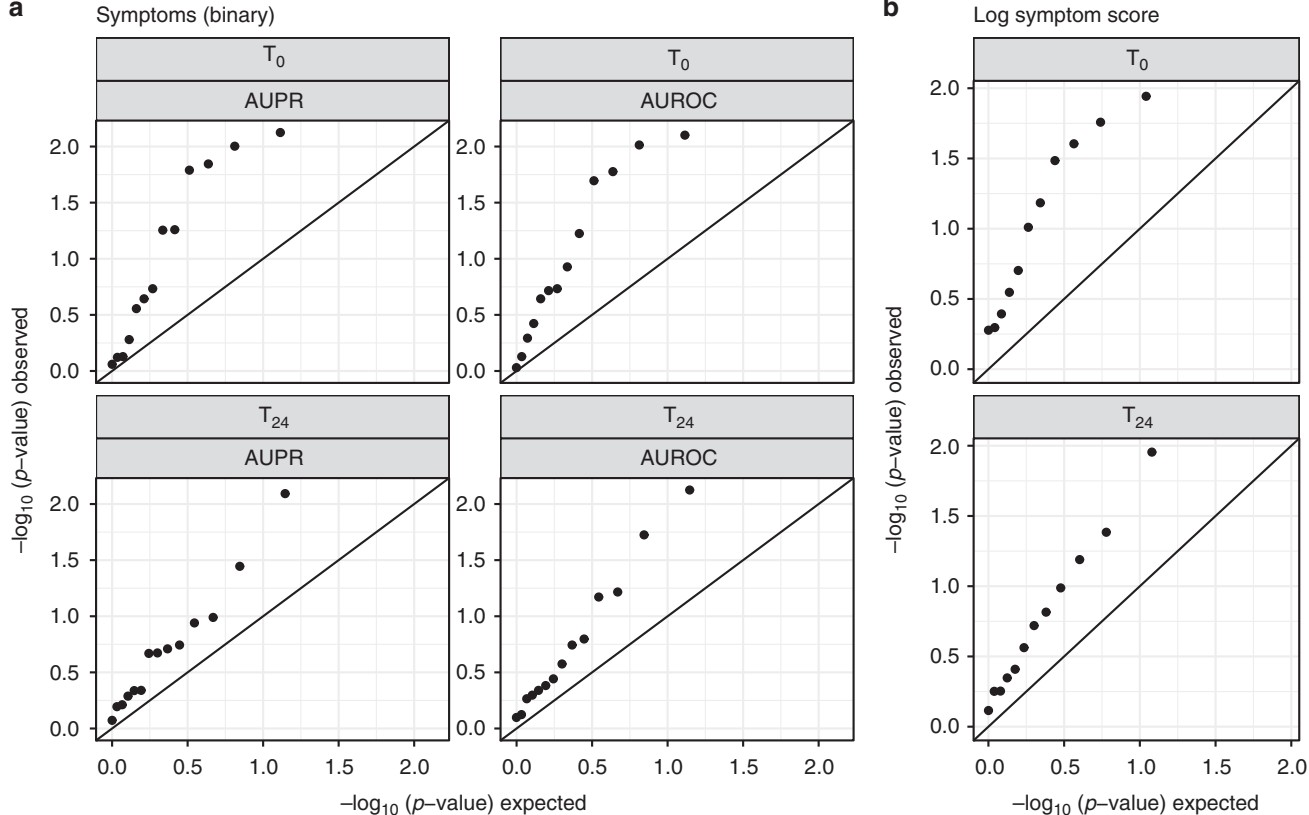

**Fig. 2** Models perform better than expected at random. Observed $-\log_{10}(p$ value) versus the null expectation for submitted predictions for predicting **a** presence of symptoms (SC2) and **b** log symptom score (SC3), where $p$ values were assessed by permutation of the predictions relative to the true values. For both subchallenges significant enrichment of $p$ values (Kolmogorov–Smirnov test for enrichment $p$ value 0.008, 0.002, 0.021, and 0.05 for AUPR($T_0$), AUROC($T_0$), AUPR($T_{24}$), and AUROC($T_{24}$), respectively, for presence of symptoms, and one-sided Kolmogorov–Smirnov test for enrichment $p$ value 0.005 and 0.035 for $T_0$ and $T_{24}$, respectively, for log symptom score) across submissions demonstrates that pre-exposure and early post-exposure transcriptomic data can predict susceptibility to respiratory viruses

that predictive models of symptoms following viral exposure can be built using pre-exposure gene expression.

Team cwruPatho constructed models of infection based on pathway modulation, rather than gene expression, to predict infection outcomes. To do so, they used a sample-level enrichment analysis (SLEA)[17] approach to summarize the expression of genes implicated in the Hallmark gene sets[18] of the Molecular Signature DataBase (MSigDB)[19]. They then fitted LASSO regularized regression models, which integrate feature selection with a regression fit[20], on the pathways to predict shedding, presence of symptoms and symptom severity following viral exposure. Their work demonstrated that including multiple genes sharing the same biological function results in more robust prediction than using any single surrogate gene.

Teams Schrodinger's Cat and cwruPatho used different feature transformation methods and machine learning approaches, suggesting that multiple approaches can successfully identify pre- or early post-exposure transcriptomic markers of viral infection susceptibility or resilience. To gauge the range of approaches taken, we extended this comparison to all Respiratory Viral DREAM Challenge teams who reported details on the methods they used to develop their submissions. We assessed the range of data preprocessing, feature selection, and predictive modeling approaches employed for the submissions, to determine whether any of these methods were associated with better prediction accuracy. Details of these three analysis steps (preprocessing, feature selection and predictive modeling) were manually extracted from reports of 24 teams (35 separate reports) who submitted predictions

either for the leaderboard test set or the independent test set. To more precisely reflect the conceptual variations across employed methodologies, each of these three analysis tasks was broken down into four data preprocessing categories, seven feature selection categories and nine predictive modeling categories (Supplementary Table 2). Twenty of 24 (83.3%) teams employed some version of data preprocessing, the task most significantly associated with predictive ability (Supplementary Figure 3A). Specifically, exclusion of sham-exposed subjects and data normalization associated best with predictive performance (Fig. 3).

Feature selection and predictive modeling approaches positively associated with predictive ability differed depending on whether the task was classification (presence of symptoms) or regression (symptom severity). Random forest-based predictive models performed slightly better than support vector machine (SVM)/support vector regression (SVR) methods at predicting symptom status (SC2) (Supplementary Figure 3B). However, there was no discernible pattern relating feature selection and improved performance in SC2. Feature selection using machine learning approaches such as cross-validation was associated with improved performance in predicting symptom severity (SC3) (Fig. 3), as were SVM/SVR approaches when compared to linear regression model-based methods (e.g. logistic regression; Supplementary Figure 3C). Of note, SVM/SVR approaches were the most popular among the submissions.

We also sought to compare cross-timepoint predictions to determine the stability of predictions by timepoint. Significant correlation was observed between predictions using $T_0$ and $T_{24}$

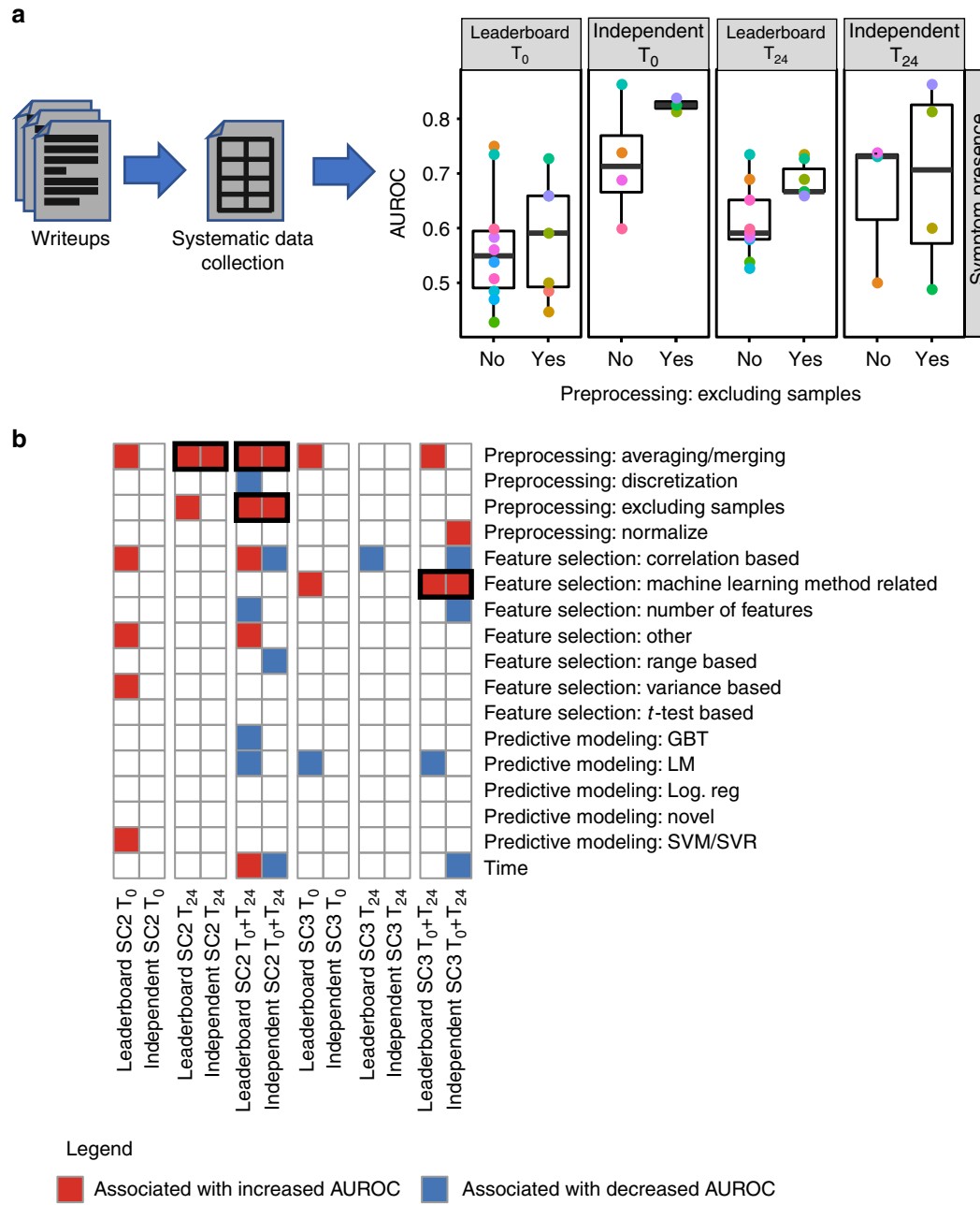

**Fig. 3** Preprocessing leads to more accurate predictions. **a** Schematic representation of the analysis of the participating teams' writeups to identify methodological steps associated with more accurate prediction of symptoms. First, the writeups were manually inspected to identify the preprocessing, feature selection and predictive modeling method used by each team. Second, the methods were regrouped into general categories across teams. Third, each general method was assessed for its association with predictive model accuracies on the leaderboard test set and the independent test set. On the boxplot, the lower whisker, the lower hinge, the mid hinge, the upper hinge and the upper whisker correspond to $-1.5\times$ IQR from the first quartile, the first quartile, the median, the third quartile and $1.5\times$ IQR from the third quartile of the AUROC, respectively. **b** Heatmap showing the association of each general method with prediction ability (i.e. AUROC for SC2 (prediction of symptom presence) and Pearson's correlation coefficient for SC3 (prediction of symptom severity)). For each general method, a Wilcoxon rank-sum test was used to assess the association between using the method (coded as a binary variable) and prediction ability

gene expression for symptomatic classification (SC2) (Leaderboard: $\rho = 0.608$, $p$ value $= 1.04\text{e-}61$; Independent test set: $\rho = 0.451$, $p$ value $= 2.05\text{e-}25$). Interestingly, we observed that approximately 25% of subjects were difficult to predict based on $T_0$ gene expression profile (inherently difficult; Supplementary Figure 4); similarly, approximately 25% of subjects were correctly predicted by the majority of teams (inherently easy; Supplementary Figure 4). Inherently difficult subjects were also misclassified when $T_{24}$ gene expression data was used for prediction. Inherently easy subjects were also consistently easy to classify

using $T_{24}$ gene expression data. This suggests ab initio characteristics allow some subjects to be more susceptible or resilient to symptomatic disease and that, within 24 h, those characteristics are not substantially altered in post-exposure peripheral blood expression profiles.

**Biological interpretation of predictors**. In addition to predictions, each team was asked to submit lists of gene expression features used in their predictive models. Twenty-four teams submitted

predictive models with AUROC > 0.5 for SC2 or $r > 0$ for SC3 (leaderboard test set) for either $T_0$ or $T_{24}$, among which six teams submitted separate models for each virus and reported virus-specific predictors. The remaining 18 reported models independent of virus, submitting a single model for all viruses. With the exception of the list from cwruPatho, which used pathway information in the selection of features, pathway analysis of individual predictor lists showed no enrichment of pathways from MSigDB[19], possibly due to the tendency of most feature selection algorithms to choose one or few features from within correlated sets.

We then assessed whether models showing predictive ability (leaderboard test set AUROC > 0.5 for SC2 or $r > 0$ for SC3) tended to pick the same gene features, or whether the different gene sets may provide complementary information. Within each subchallenge and timepoint, the significance of the overlap among predictor lists was calculated for every combination of two or more predictor lists across teams. All two-way, three-way, four-way, etc. overlaps were considered. This analysis revealed that there were no genes shared among all teams for any timepoint or subchallenge (Fig. 4a).

Despite the paucity of overlap among predictor lists, we sought to identify whether genes used in the predictive models were part of the same biological processes or pathways. In other words, we examined whether different teams might have chosen different surrogate genes to represent the same pathway. To test this hypothesis, we performed pathway enrichment analysis of the union of predictors across predictor lists within timepoint and subchallenge. We observed significant enrichments in each case (Fig. 4b), suggesting that predictive gene features are indeed complementary across models. More pathways were enriched among predictors from $T_{24}$ models (SC2 = 17 pathways and SC3 = 20 pathways) than from $T_0$ models (SC2 = 15 pathways and SC3 = 17 pathways). At $T_0$, genes involved in the metabolism of heme and erythroblast differentiation (heme metabolism), genes specifically upregulated by KRAS activation (KRAS signaling (up)), genes defining an inflammatory response (inflammatory response) and genes mediating cell death by activation of caspases (apoptosis) were associated with presence of symptoms in both SC2 and SC3 (Fig. 4b). At $T_{24}$, along with heme metabolism, the expression of several inflammatory response pathways like KRAS signaling, inflammatory response, genes upregulated in response to the gamma cytokine IFNg (interferon gamma response), genes upregulated by IL6 via STAT3 (IL6 JAK STAT3 signaling), genes regulated by NF-κB in response to TNF (TNFA signaling via NFKB) and genes encoding components of the complement system (complement) were associated with symptoms in both SC2 and SC3 (Fig. 4b). Additionally, there was a significant overlap in genes across timepoints and subchallenges in each of these enriched pathways (Fisher's exact test $p$ value ≤ 0.05) (Supplementary Data 2).

A meta-analysis across subchallenges (SC2 and SC3) and timepoints ($T_0$ and $T_{24}$) was performed in order to identify the most significant pathways associated with outcome. Heme metabolism was the most significantly associated with developing symptoms (susceptibility), while oxidative phosphorylation and MYC targets were the most significantly associated with a lack of symptoms (resilience) (Supplementary Figure 5). This indicates that heme, known to generate inflammatory mediators through the activation of selective inflammatory pathways[21] is the best predictor of becoming symptomatic both pre- and early post-exposure to respiratory viruses. Genes in heme metabolism associated with symptoms include genes coding for the hemoglobin subunits (HBB, HBD, HBQ1, and HBZ), the heme binding protein (HEBP1) and genes coding for enzymes important for the synthesis of heme (ALAS2, FECH, HMBS, UROD). It also includes glycophorins, which are the major erythrocyte membrane proteins (GYPA, GYPB,

GYPC, and GYPE), which are known receptors for the influenza virus (Fig. 4c)[22,23]. Genes essential for erythroid maturation and differentiation (NEF2, TAL1, EPOR, and GATA1), including the transcription factor GATA1 and its targets, the hemoglobin subunit genes HBB and HBG1/2, were also part of heme metabolism associated with an increase in symptom frequency and severity.

## Discussion

Using an open data analysis challenge framework, this study showed that models based on transcriptomic profiles, even prior to viral exposure, were predictive of infectious symptoms and symptom severity, which has not been previously demonstrated. The best scoring individual models for predicting symptoms and log-symptom score, though statistically significant, fall short of practical significance. However, these outcomes suggest that there is potential to develop models and ultimately, clinically relevant tests, based on the knowledge gained from these results. This would necessitate further efforts to generate more data or identify different biomarker assays which more accurately assess the mechanisms observed in the transcriptomic models. Additionally, since these studies focused on healthy adults, further data generation should extend to a wider range of subjects with respect to age and health status, as well as tracking and modeling these cofactors.

A generally useful exercise in crowdsourcing-based challenges is to construct ensembles from the submissions to assimilate the knowledge contained in them, and boost the overall predictive power of the challenge[24]. This exercise has yielded useful results in earlier benchmark studies[25,26] and the DREAM Rheumatoid Arthritis Challenge[27]. However, the ensembles constructed for the Respiratory Viral DREAM Challenge did not perform better than the respective best performers among all the individual submissions for the various subchallenges and timepoints. We attribute this shortcoming partly to the relatively small training set (118 subjects), which may incline the ensemble methods to overfit these data, and the assumption of class-conditioned independence of the submissions inherent in SUMMA may not have been appropriate in this challenge[28]. The relative homogeneity, or lack of diversity, among the submissions for the various subchallenges and timepoints may have been another potential factor behind the diminished performance of the ensembles[29].

The relative homogeneity of submissions and observation that the same subjects are misclassified by almost all participating teams suggests there may be a plateau in predictive ability when using gene expression to predict the presence of symptoms or symptom severity. It is possible that an integrative analysis supplementing or replacing the gene expression data with post-transcriptional (such as metabolomic or proteomic) data could further improve accuracy. For example, metabolomic data have been used to differentiate patients with influenza H1N1 from others with bacterial pneumonia or non-infectious conditions as well as differentiate influenza survivors from nonsurvivors[30]. With respect to proteomics, Burke et al. used four of the viral exposure studies described here to derive and validate a proteomic signature from nasal lavage samples which distinguish, with high accuracy, symptomatic from asymptomatic subjects at the time of maximal symptoms[31]. Several cytokines have been investigated in a variety of infectious disease conditions. Of particular relevance, cytokine profiling has been performed for one of the influenza H3N2 studies used in this Challenge. In that work, McClain et al. demonstrated that several cytokines were upregulated early after viral exposure (within 24 h in some cases) and differentiated symptomatic from asymptomatic cases[32]. Baseline differences in cytokine expression were not observed, however, suggesting that cytokine expression is useful for predicting response to viral exposure but not baseline susceptibility. To our knowledge, no study has identified baseline metabolomic or proteomic

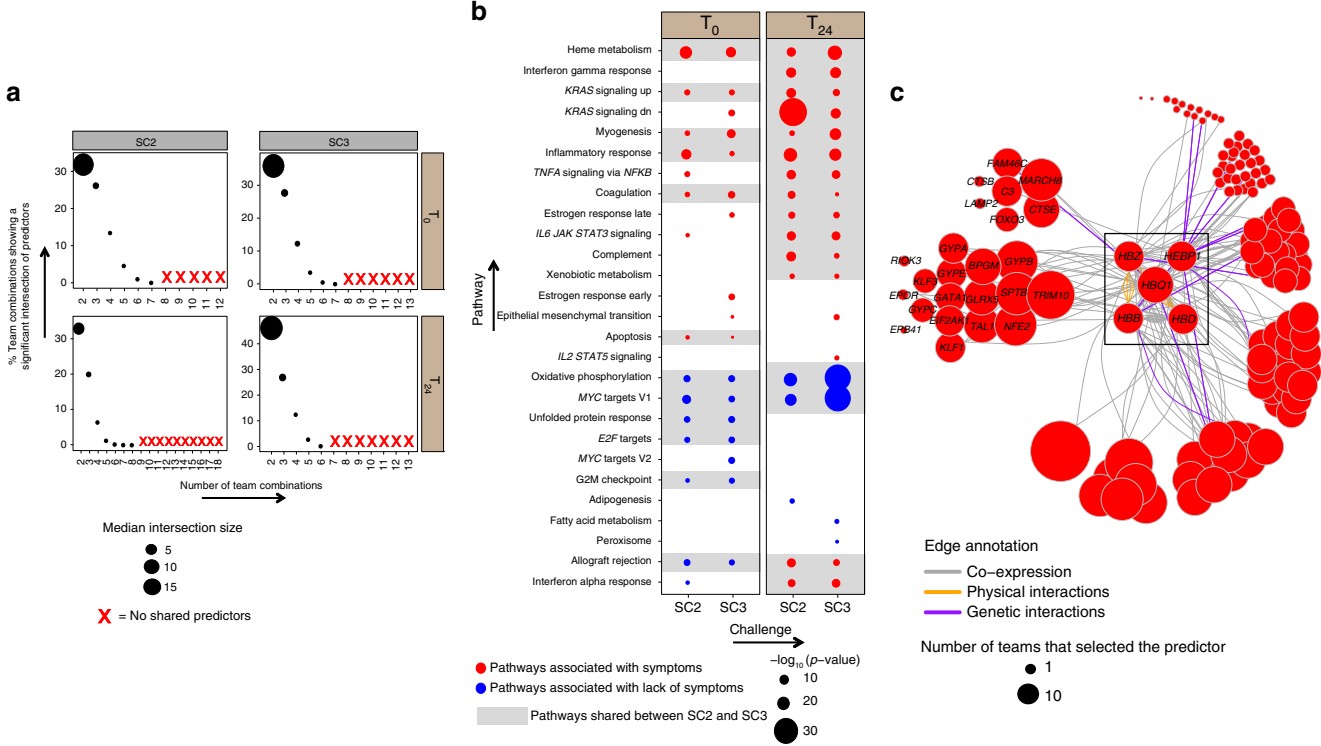

**Fig. 4** Overlap and pathway enrichment among predictors of symptoms. **a** Percent of team combinations showing statistically significant intersections of predictors at $T_0$ and $T_{24}$. Only teams with AUROC $\geq 0.5$ or $r \geq 0$ for subchallenges 2 and 3, respectively, were used for this analysis. The x-axis indicates the number of teams included in the combination. For example, the value 2 corresponds to pairwise overlaps, 3 corresponds to 3-way overlaps, etc. The y-axis indicates the percentage of team combinations with a statistically significant (p value < 0.05) predictor intersection. Point size indicates median intersection size of predictors among team combinations with significant predictor intersection; "X" indicates no significant predictor intersection. **b** Pathway enrichment among predictors of infection for each subchallenge (SC2 and SC3) at $T_0$ and $T_{24}$. The x-axis indicates subchallenge and each grid indicates timepoint. The y-axis indicates pathways enriched among predictors with a Benjamini−Hochberg-corrected Fisher's exact test p value < 0.05. Point size represents the Fisher's exact test enrichment $-\log_{10}(p$ value). Point colors indicate whether the pathway was associated with symptoms (red) or lack thereof (blue). Pathways shared between both SC2 and SC3 at each timepoint are highlighted in gray. Pathways are ordered by the decreasing maxP test statistic as determined in Supplementary Figure 5. **c** GeneMANIA network of the union of predictors involved in the Heme metabolism pathway across timepoints ($T_0$ and $T_{24}$) and subchallenges (SC2 and SC3). Edges are inferred by GeneMANIA[51] corresponding to coexpression (gray), physical interactions (orange), and genetic interactions (purple) among genes. Node size corresponds to the number of teams that selected the predictor

predictors of resilience or susceptibility to respiratory viral infection. In addition, the combination of these data with transcriptomic predictors has not yet been investigated and may yield robust predictors of susceptibility or resistance to infection.

Our analyses revealed a significant concordance between predictions at $T_0$ and $T_{24}$ (Supplementary Figure 4), as well as a significant overlap between predictors at each of these timepoints (Supplementary Data 2). Given the stability of predictions and predictors between $T_0$ and $T_{24}$, it appears that the pre-exposure biological mechanisms conferring susceptibility or resilience to respiratory viral infection may be observable up to 1 day post-exposure. We also observed significant overlap between gene signatures at both $T_0$ and $T_{24}$ and late stage signatures of viral infection, reported in the literature, and derived from gene expression 48 h or later after viral exposure (Supplementary Data 3)[5–9,15,33–38]. The overlap between the predictors identified in this study and the later stage signatures was more significant at $T_{24}$ than $T_0$, suggesting that pre-exposure signatures of susceptibility differ somewhat from post-exposure signatures of active infection, and $T_{24}$ predictors may reflect some aspects of both. The $T_0$ gene signatures may encompass novel insight into ab initio factors that confer resilience or susceptibility.

Pathway enrichment analysis in our study revealed that the most significantly enriched pathway associated with symptomatic infection was heme metabolism, known to have a direct role in

immunity through activation of innate immune receptors on macrophages and neutrophils[21]. Of note, genes part of heme metabolism were also enriched among late stage signatures of viral infection (ex. Hemoglobin gene *HBZ* and the iron containing glycoprotein *ACP5* in ref.[33]). Iron (obtained from heme) homeostasis is an important aspect of human health and disease. Viruses require an iron-rich host to survive and grow, and iron accumulation in macrophages has been shown to favor replication and colonization of several viruses (e.g. HIV-1, HCV) and other pathogenic microorganisms[39]. Furthermore, iron-replete cells have been shown to be better hosts for viral proliferation[39]. Increased iron loading in macrophages positively correlates with mortality[39] and it has been shown that viral infection can cause iron overload which could further exacerbate disease. Additionally, previous evidence suggests counteracting iron accumulation may limit infection[21,39]. Studies have shown that limiting iron availability to infected cells (by the use of iron chelators) curbed the growth of several infectious viruses and ameliorated disease[21,39–41]. This important role of iron in the susceptibility and response to infection may be the mechanism by which heme metabolism genes conferred susceptibility to respiratory viral infection. As such, it represents an important biological pathway potentially offering a means by which an individual's susceptibility or response to infection can be optimized. Such a relationship should be investigated in future studies of infection susceptibility. In addition, Heme-oxygenase (*HMOX1*), a

heme-degrading enzyme that antagonizes heme-induced inflammation and is essential for the clearance of heme from circulation[42], was among the predictors from the $T_0$ models. Interestingly, the expression of this gene at baseline was associated with a lack of symptoms (for both SC2 and SC3), in concordance with its reported antiviral role during influenza infection[43,44]. Augmentation of *HMOX1* expression by gene transfer had provided cellular resistance against heme toxicity[45]. Hence enhancing *HMOX1* activity could be an alternative to antagonize heme-induced effects and thereby controlling infection and inflammation.

In addition to heme metabolism, pro-inflammatory pathways such as inflammatory response, *KRAS* signaling, and apoptosis were also associated with susceptibility to viral infection in our study, while homeostatic pathways, such as oxidative phosphorylation and *MYC* targets, were associated with resilience, both prior to and post viral exposure (Fig. 4). Enrichment of these pathways among $T_{24}$ predictors was more significant than among the $T_0$ predictors, suggesting these mechanisms are not only emblematic of baseline system health, but also response to viral invasion. Additional pathways enriched among $T_{24}$ predictors include interferon gamma response and complement, which are involved in innate and acquired immunity. Several genes among $T_0$ and $T_{24}$ predictors overlapped with genes positively associated with flu vaccination response[46]. Among them, *FCER1G* and *STAB1*, members of the inflammatory response pathway positively associated with symptoms in this study and were elevated prior to vaccination in young adults who showed good response to vaccination[46] (Fisher exact test: $p = 0.0338$ for $T_0$ and $p = 0.000673$ for $T_{24}$). This suggests that individuals predicted at a higher risk of presenting symptoms following influenza exposure may also be the most likely to benefit from vaccination.

The Respiratory Viral DREAM Challenge is to date the largest and most comprehensive analysis of early stage prediction of viral susceptibility. The open data analysis challenge framework is useful for comparing approaches and identifying the most scientifically or clinically relevant model or method in an unbiased fashion[24]. In this case, we observed few commonalities among the best performing models of symptomatic susceptibility to respiratory viral exposure. Indeed, the overall best performing teams in the challenge used different machine learning techniques to build their models. Interestingly, data preprocessing was the analysis task most significantly associated with model accuracy, suggesting what has often been speculated, that adequate attention to data processing prior to predictive modeling is a crucial first step[47].

The open data challenge framework is also useful in arriving at consensus regarding research outcomes that may guide future efforts within a field[24]. Through this challenge, we have identified ab initio transcriptomic signatures predictive of response to viral exposure, which has provided valuable insight into the biological mechanisms conferring susceptibility to infection. This insight was not evident from any individual model, but became apparent with the meta-analysis of the individual signatures. While development of a diagnostic test of baseline susceptibility is not yet feasible based on these findings, they suggest potential for development in this area.

## Methods

**Training data.** Training data came from seven related viral exposure trials, representing four different respiratory viruses. The data sets are DEE1 RSV, DEE2 H3N2, DEE3 H1N1, DEE4X H1N1, DEE5 H3N2, Rhinovirus Duke, and Rhinovirus UVA[6,7,9]. In each of these human viral exposure trials, healthy volunteers were followed for 7−9 days following controlled nasal exposure to the specified respiratory virus. Subjects enrolled into these viral exposure experiments had to meet several inclusion and exclusion criteria. Among them was an evaluation of pre-existing neutralizing antibodies to the viral strain. In the case of influenza H3N2 and influenza H1N1, all subjects were screened for such antibodies. Any subject with pre-existing antibodies to the viral strain was excluded. For the

rhinovirus studies, subjects with a serum neutralizing antibody titer to RV39 > 1:4 at prescreening were excluded. For the RSV study, subjects were prescreened for neutralizing antibodies, although the presence of such antibodies was not an exclusion criterion.

Symptom data and nasal lavage samples were collected from each subject on a repeated basis over the course of 7−9 days. Viral infection was quantified by measuring release of viral particles from nasal passages (viral shedding), as assessed from nasal lavage samples via qualitative viral culture and/or quantitative influenza RT-PCR. Symptom data were collected through self-report on a repeated basis. Symptoms were quantified using a modified Jackson score[14], which assessed the severity of eight upper respiratory symptoms (runny nose, cough, headache, malaise, myalgia, sneeze, sore throat, and stuffy nose) rated 0−4, with 4 being most severe. Scores were integrated daily over 5-day windows.

Blood was collected and gene expression of peripheral blood was performed 1 day (24−30 h) prior to exposure, immediately prior to exposure, and at regular intervals following exposure. These peripheral blood samples were gene expression profiled on the Affy Human Genome U133A 2.0 array.

All subjects exposed to influenza (H1N1 or H3N2) received oseltamivir 5 days post-exposure. However, 14 (of 21) subjects in the DEE5 H3N2 cohort received early treatment (24 h post-exposure) regardless of symptoms or shedding. Rhinovirus Duke additionally included seven volunteers who were exposed to sham rather than active virus.

All subjects provided written consents, and each of the seven trials was reviewed and approved by the appropriate governing IRB.

**RSV test data.** Healthy nonsmoking adults aged 18−45 were eligible for inclusion after screening to exclude underlying immunodeficiencies. A total of 21 subjects (10 female) were inoculated with $10^4$ plaque-forming units of RSV A Memphis 37 (RSV M37) by intranasal drops and quarantined from 1 day before inoculation to the 12th day after. Peripheral blood samples were taken immediately before inoculation and regularly for the next 7 days and profiled on the Affy Human Genome U133A 2.0 array. Subjects were discharged after study day 12, provided no or mild respiratory symptoms and a negative RSV antigen respiratory secretions test. Shedding was determined by polymerase chain reaction (PCR) in nasal lavage and defined as detectable virus for ≥2 days between day +2 and day +10 to avoid false-positives from the viral inoculum and to align case definitions with the other seven studies. Subjects filled a diary of upper respiratory tract symptoms from day −1 to day +12, which was summarized using a modified Jackson score. All subjects returned for further nasal and blood sampling on day +28 for safety purposes. All subjects provided written informed consent and the study was approved by the UK National Research Ethics Service (London-Fulham Research Ethics Committee ref. 11/LO/1826).

**Gene expression normalization.** Both raw (CEL files) and normalized versions of the gene expression data were made available to teams in the Challenge. Both versions contained only profiles that pass QC metrics including those for RNA Degradation, scale factors, percent genes present, β-actin 3′ to 5′ ratio and GAPDH 3′ to 5′ ratio in the Affy Bioconductor package. Normalization via RMA was performed on all expression data across all timepoints for the training and leaderboard data sets. The RSV data were later normalized together with the training and leaderboard data, and teams were free to further QC and normalize the data in the way they deemed appropriate.

**Analysis challenge design.** The training data studies were split into training and leaderboard sets, where the leaderboard subjects were chosen randomly from three of the trials: DEE4X H1N1, DEE5 H3N2, and Rhinovirus Duke, which were not publicly available at the time of challenge launch. Outcome data for the leaderboard set were not provided to the teams, but instead, teams were able to test predictions in these individuals using the leaderboard, with a maximum of six submissions per subchallenge, the purpose of which was to allow teams to optimize their models prior to assessment on the independent test data. Of these, at least one submission was required to use only data prior to viral exposure and at least one using data up to 24 h post-exposure.

For the training data, teams had access to clinical and demographic variables: age, sex, whether the subject received early oseltamivir treatment (DEE5 H3N2 only) and whether the subject received sham exposure rather than virus (Rhinovirus Duke only), as well as gene expression data for the entire time-course of the studies. They also received data for the three outcomes used in the data analysis challenge:

- Subchallenge 1: SHEDDING_SC1, a binary variable indicating the presence of virus in nasal swab following exposure;
- Subchallenge 2: SYMPTOMATIC_SC2, a binary variable indicating post-exposure maximum 5-day integrated symptom score ≥6;
- Subchallenge 3: LOGSYMPTSCORE_SC3, a continuous variable indicating the log of the maximum 5-day integrated symptom score +1

as well as the granular symptom data by day and symptom category. For the leaderboard test data, they were supplied with the clinical and demographic variables and gene expression data up to 24 h post-exposure.

Final assessment of optimized models was performed in the RSV Test Data (i.e. the independent test set), and outcomes for these subjects were withheld from teams. In order to assure that predictions were limited to data from the appropriate time window, the gene expression data were released in two phases corresponding to data prior to viral exposure, and data up to 24 h post exposure. Teams were also supplied with age and sex information for these subjects.

The Challenge was launched and training data were released on May 15, 2016 for participants to use to begin analyzing the data and building their models. In total 38 teams registered for the challenge and 37 participated (Supplementary Table 1). The leaderboards opened approximately 2 months later, and were open for approximately 3 months (July to September) to allow participants to optimize their models with feedback from the scores on the leaderboard data. At the close of this round on September 30, participating teams were also required to submit code, methodological writeups, predictor lists, and LOOCVs, and doing so qualified participants to be included as authors (either Consortium or by-line) on this manuscript. Participating teams could opt to evaluate their optimized models in the independent test data, which occurred from January to February 2017. At the close of the challenge, participating teams were invited to collaborate with the Challenge Organizers to analyze the results. Prior to the launch of the challenge, substantial effort was put forth by the Challenge organizers to collate and vet the data, to determine the feasibility of the Challenge and define the Challenge objectives. For further details on the organizational efforts required to prepare for a challenge, see Saez-Rodriguez et al.[24].

**Submission scoring**. Team predictions were compared to true values using AUPR and AUROC for subchallenges 1 and 2, and Pearson correlation for subchallenge 3. For each submission, a $p$ value, estimating the probability of observing the score under the null hypothesis that the predicted labels are random, was computed by 10,000 permutations of the predictions relative to the true values.

Enrichment of $p$ values of the submitted models was assessed via 1-sided Kolmogorov–Smirnov test with a null hypothesis that the $p$ values follow a U[0,1] distribution, and an alternative hypothesis that they follow a distribution that is stochastically smaller than U[0,1].

**Heterogeneity of the predictions**. $T_0$ and $T_{24}$ predictions for each outcome and team were collected to assess whether they were correlated. Three teams provided predictions as binary values while 12 teams provided predictions as continuous values on different scales. In order to compare binary and continuous predictions, we first transformed them into ranks (with ties given the same average rank) and then ordered subjects increasingly by their mean rank across outcomes (mean-rank). The lower the mean-rank, the more likely a subject was predicted by the teams as not showing symptoms, whereas a higher mean-rank means a subject was predicted by most of the teams as showing symptoms. Distribution of the mean-rank (Supplementary Figure 4) revealed three groups of subjects: (1) ~25% of subjects correctly predicted by most of the teams (i.e. inherently easy), (2) ~25% of subjects incorrectly predicted by most of the teams (i.e. inherently difficult) and (3) ~50% of subjects who were predicted differently by the teams.

**Ensemble prediction**. We constructed a variety of ensembles from the teams' submissions to the various subchallenges as a part of the collaborative phase of the Respiratory Viral DREAM Challenge. To enable a comparative analysis between individual and ensemble models in the collaborative phase, the teams were requested to submit LOOCV-derived predictions on the training examples using the same methods used to generate leaderboard and/or test set predictions in the competitive phase. The LOOCV setup, which does not involve random subsetting of the training data, was chosen to avoid potential overfitting that can otherwise occur from training and testing on predictions made on the same set of examples[25]. We used three types of approaches for learning ensembles, namely stacking and its clustering-based variants[25], Reinforcement Learning-based ensemble selection[26] methods, as well as SUMMA, an unsupervised method for the aggregation of predictions[28]. Consistent with the process followed by the individual teams, we learned all the ensembles using the training set LOOCV-derived predictions described above, and used the leaderboard data to select the final models to be evaluated on the test data.

**Combined gene sets**. Statistical significance of the overlap among predictor lists was calculated using the multiset intersection probability method implemented in the SuperExactTest R package[48]. A first set of analysis was performed with teams whose leaderboard AUROC > 0.5. A second set of analysis aimed at identifying genes that overlap virus-specific, subchallenge-specific and timepoint-specific predictive models, and was restricted to teams that provided virus-specific (Nautilus, aydin, SSN_Dream_Team, Txsolo, cwruPatho and Aganita), subchallenge-specific (aydin, SSN_Dream_Team, cwruPatho, jhou) and timepoint-specific predictors (aydin, SSN_Dream_Team, cwruPatho, Espoir, jdn, jhou, burkhajo) and participated in the leaderboard phase of the challenge, respectively. For both analyses, overlapping predictors associated with $p$ values less than or equal to 0.005 were considered significant[49].

**Pathway enrichment analysis**. To assess pathway enrichment among predictors of infection, we considered predictors from teams with leaderboard AUROC > 0.5 (SC2) or Pearson correlation, $r > 0$ (SC3). Affymetrix Human U133A 2.0 GeneChip probe identifiers were mapped to gene symbols. We removed probes matching multiple genes, and when multiple probes matched a single gene, we retained the probe with the maximum median intensity across subjects.

For the list of predictors of presence of symptoms (SC2), we calculated the log2 fold-change of features (symptomatic(1)/asymptomatic(0)) at $T_0$ and $T_{24}$, and for prediction of the symptom scores (SC3), we calculated the Spearman's correlation coefficient of the features, at $T_0$ and $T_{24}$, with the outcome. Pathway enrichment was then performed on the union of all predictors (across the teams) that were associated with presence/increase severity of symptoms (SC2: log2 fold-change > 0 and SC3: Spearman's correlation > 0), as well as, for the union of all predictors (across teams) that were associated with lack of symptoms/lower symptoms severity (SC2: log2 fold-change < 0 and SC3: Spearman's correlation < 0), separately by timepoint and subchallenge. We used the Hallmark gene sets (version 6.0)[18] of the Molecular Signature DataBase (MSigDB)[19] for the enrichment analysis, and calculated the significance using Fisher's exact test. The resulting $p$ values were corrected for multiple comparisons using the Benjamini and Hochberg algorithm. Only significantly enriched pathways (corrected $p$ value < 0.05) were reported. Meta-analyses across subchallenges and timepoints were performed using the maxP test statistic[50].

**Code availability**. Code for individual models are available through www.synapse.org/ViralChallenge.

## Data availability

Data are available through GEO GSE73072. Challenge results and methods and code for individual models are available through www.synapse.org/ViralChallenge. The authors declare that all other data supporting the findings of this study are available within the article and its Supplementary Information files, or are available from the authors upon request.

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

## Acknowledgements

This work was supported by Defense Advanced Research Projects Agency and the Army Research Office through Grant W911NF-15-1-0107. The views expressed are those of the authors and do not reflect the official policy or position of the Department of Defense or the U.S. Government. J.G.B. was supported by a training grant from the National Institutes of Health, USA (NIH grant 4T15LM007088-25). G.P. and A.S.'s work was supported by NIH grant # R01GM114434 and an IBM faculty award to G.P. T.E.M.N. was supported by the Ministry of Science and Technology of Taiwan grants MOST 105-2218-E-006-016-MY2 and 107-2634-F-006-009. K.Y.Y. was supported by NIH grants U54 HL127624 and R01GM126019. M.S. was supported by Grants-in-Aid for Scientific Research JP16H02866 from the Japan Society for the Promotion of Science. We wish to thank the DARPA Biochronicity program and its program manager, Dr. Jim Gimlett, for generously offering to share gene expression data generated as part of that program and Rafick P. Sekaly (Case Western Reserve University) for his critical feedback during the writing process.

## Author contributions

R.H., C.C., M.T.M., C.W.W., G.S.G., and E.L.T. devised and performed the viral exposure experiments. R.H., T.Y., G.S.G., E.L.T., L.M.M., and S.K.S. designed and ran the data analysis challenge. S.F., A.T., M.M., J.G.B., R.K., Z.A., K.Y.Y., R.A., S.J., X.L., T.E.M.N., M.S., L.L.E., and The Respiratory Viral DREAM Challenge Consortium members participated in the Challenge and S.F., A.T., M.M., J.G.B., R.K., R.H., Z.A., K.Y.Y., M.E.A., R.A., S.J., X.L., T.E.M.N., M.S., A.S., R.V., G.P., L.L.E., and S.K.S. analyzed the data.

## Additional information

**Competing interests:** E.L.T. reports personal fees from bioMerieux. E.L.T., C.W.W., R.H. and M.T.M. report grants from NIAID. E.L.T., C.W.W., R.H., M.T.M., and G.S.G. report grants from DARPA. E.L.T., C.W.W., and G.S.G. are founders and have equity in Predigen, Inc. E.L.T., M.T.M., C.W.W., R.H., and G.S.G. have a patent pending for methods to diagnose and treat acute respiratory infections. The remaining authors declare no competing interests.

## The Respiratory Viral DREAM Challenge Consortium

Emna Ben Abdallah[21,22], Farnoosh Abbas Aghababazadeh[23], Alicia Amadoz[24], Sherry Bhalla[25], Kevin Bleakley[26,27], Erika Bongen[28], Domenico Borzacchiello[22,29], Philipp Bucher[30,31], Jose Carbonell-Caballero[32], Kumardeep Chaudhary[33], Francisco Chinesta[34], Prasad Chodavarapu[35], Ryan D Chow[36], Thomas Cokelaer[37], Cankut Cubuk[38], Sandeep Kumar Dhanda[39], Joaquin Dopazo[38], Thomas Faux[2], Yang Feng[40], Christofer Flinta[41], Carito Guziolowski[21,22], Di He[42], Marta R. Hidalgo[38], Jiayi Hou[43], Katsumi Inoue[44,45], Maria K Jaakkola[2,46], Jiadong Ji[47], Ritesh Kumar[48], Sunil Kumar[30,31], Miron Bartosz Kursa[49], Qian Li[50,51], Michał Łopuszyński[49], Pengcheng Lu[51], Morgan Magnin[21,22,44], Weiguang Mao[52,53], Bertrand Miannay[21], Iryna Nikolayeva[54,55,56], Zoran Obradovic[57], Chi Pak[58], Mohammad M. Rahman[10], Misbah Razzaq[21,22], Tony Ribeiro[21,22,44], Olivier Roux[21,22], Ehsan Saghapour[59], Harsh Saini[60], Shamim Sarhadi[61], Hiroki Sato[62], Benno Schwikowski[54], Alok Sharma[63,64,65], Ronesh Sharma[65,66], Deepak Singla[67], Ivan Stojkovic[57,68], Tomi Suomi[2], Maria Suprun[69], Chengzhe Tian[70,71], Lewis E. Tomalin[72], Lei Xie[73] & Xiang Yu[74]

[21]Laboratoire des Sciences du Numérique de Nantes, 44321 Nantes, France. [22]École Centrale de Nantes, 44321 Nantes, France. [23]Department of Biostatistics and Bioinformatics, Moffitt Cancer Center, Tampa, FL 33612, USA. [24]Department of Bioinformatics, Igenomix SL, 46980 Paterna, Spain. [25]CSIR-Institute of Microbial Technology, Chandigarh 160036, India. [26]Inria Saclay, 91120 Palaiseau, France. [27]Département de Mathématiques d'Orsay, 91405 Orsay, France. [28]Stanford Immunology, Stanford, CA 94305, USA. [29]Institut de Calcul Intensif, 44321 Nantes, France. [30]Swiss Institute for Experimental Cancer Research, Swiss Federal Institute of Technology Lausanne (EPFL), 1015 Lausanne, Switzerland. [31]Swiss Institute of Bioinformatics, 1015 Lausanne, Switzerland. [32]Centre de Regulacio Genomica (CRG), Barcelona Institute for Science and Technology, 09003 Barcelona, Spain. [33]Epidemiology Program, University of Hawaii Cancer Center, Honolulu, HI 96813, USA. [34]PIMM, ENSAM ParisTech, 75013 Paris, France. [35]Aganitha Cognitive Solutions, S.R. Shetty Nagar, Bangalore 560 076, India. [36]Department of Genetics, Yale School of Medicine, New Haven, CT 06510, USA. [37]Institut Pasteur—Bioinformatics and Biostatistics Hub—C3BI, USR3756 IP CNRS, Paris 75015, France. [38]Clinical Bioinformatic Area, Fundacion Progreso y Salud, 41012 Sevilla, Spain. [39]Division of Vaccine Discovery, La Jolla Institute for Allergy and Immunology, La Jolla, CA 92037, USA. [40]Department of Statistics, Columbia University, New York, NY 10027, USA. [41]Ericsson Research, Machine Intelligence and Automation, 164 83 Stockholm, Sweden. [42]Department of Computer Science, Graduate Center, The City University of New York, New York, NY 10016, USA. [43]Altman Translational and Clinical Research Institute, University of California, San Diego, La Jolla, CA 92037, USA. [44]National Institute of Informatics, Chiyoda-ku, Tokyo 101-8430, Japan. [45]Tokyo Institute of Technology, Meguro-ku, Tokyo 152-8550, Japan. [46]Department of Mathematics and Statistics, University of Turku, FI-20014 Turku, Finland. [47]Department of Mathematical Statistics, School of Statistics, Shandong University of Finance and Economics, 250014 Jinan, Shandong, China. [48]CSIR-Central Scientific Instruments Organization, Chandigarh 160030, India. [49]Interdisciplinary Centre for Mathematical and Computational Modelling, University of Warsaw, 02-106 Warsaw, Poland. [50]Health Informatics Institute, Morsani College of Medicine, University of South Florida, Tampa, FL 33620, USA. [51]Department of Biostatistics, University of Kansas Medical Center, Kansas City, KS 66160, USA. [52]Department of Computational and Systems Biology, School of Medicine, University of Pittsburgh, Pittsburgh, PA 15260, USA. [53]Carnegie Mellon-University of Pittsburgh, Pittsburgh, PA 15260, USA. [54]Systems Biology Laboratory, Center for Bioinformatics, Biostatistics, and Integrative Biology (C3BI) and USR 3756, Institut Pasteur, 75015 Paris, France. [55]Unité de Génétique fonctionnelle des maladies infectieuses, Institut Pasteur, 75015 Paris, France. [56]Université Paris-Descartes, Sorbonne Paris Cité, Paris 75014, France. [57]Center for Data Analytics and Biomedical Informatics, College of Science and Technology, Temple University, Philadelphia, PA 19122, USA. [58]UT Southwestern Medical Center at Dallas, Dallas, TX 75390, USA. [59]Department of Biomedical Engineering, School of Advanced Technologies in Medicine, Isfahan University of Medical Sciences, Isfahan 8174673461, Iran. [60]Research Innovation and International, University of the South Pacific, Suva, Fiji. [61]Department of Medical Biotechnology, Faculty of Advanced Medical Sciences, Tabriz University of Medical Sciences, Tabriz 51368, Iran. [62]Graduate School of Natural Science and Technology, Gifu University, Gifu 501-1193, Japan. [63]Laboratory of Medical Science Mathematics, RIKEN Center for Integrative Medical Science, Yokohama 230-0045, Japan. [64]Institute for Integrated and Intelligent Systems, Griffith University, Brisbane, QLD 4111, Australia. [65]School of Engineering and Physics, Faculty of Science Technology and Environment, University of the South Pacific, Suva, Fiji. [66]School of Electrical and Electronics Engineering, Fiji National University, Suva, Fiji. [67]Host—Parasite Interaction Biology Group, National Institute of Malaria Research, New Delhi 110077, India. [68]Signals and Systems Department, School of Electrical Engineering, University of Belgrade, 11120 Belgrade, Serbia. [69]Department of Pediatrics, Allergy and Immunology, Icahn School of Medicine at Mount Sinai, New York, NY 10029, USA. [70]Niels Bohr Institute, University of Copenhagen, 2100 Copenhagen, Denmark. [71]Department of Chemistry and Biochemistry, University of Colorado, Boulder, Boulder, CO 80303, USA. [72]Icahn School of Medicine at Mount Sinai, New York, NY 10029, USA. [73]Department of Computer Science, The City University of New York, New York, NY 10065, USA. [74]Department of Biology, University of Pennsylvania, Philadelphia, PA 19104, USA

