## [Peer Review File · Nature Communications]

Reviewers' comments:

Reviewer #1 (Remarks to the Author):

The paper is well written and an interesting read. I have many difficulties in justifying its publication in this journal. It should be posted back to the challenge pages.

It comes to basically to the same conclusion that the Dream challenge rheumatoid arthritis proposed and published in this journal in 2016. We need to have multiomic datasets of sufficient size to probably achieve any meaningful conclusions.

In this particular case of respiratory viruses disease severity, no commonalities were found in the models produced. The level of prediction is not clinically useful. The problem is complex, RSV is mostly children and Influenza mostly older adults. We should include co-morbidities, age, respiratory capabilities, metabolomic data, among others important modulators of disease severity.

The links to the HEME pathway are interesting but remain speculative.

Reviewer #2 (Remarks to the Author):

This manuscript reports results of a modeling challenge aiming at the identification of blood transcriptional gene signatures predictive of outcomes of exposure to respiratory viruses.

The initiative is excellent and the approach seems well principled. However, some points need to be clarified or addressed.

The overall approach is interesting and appears to have been well thought out but it is also somewhat intricate for outsiders. A figure presenting the overall workflow/process would be helpful.

Some background on how such challenges are prepared and run in general and this one, in particular, would probably be useful as well. It is well explained on the website but could be at least summarized in the manuscript. Some context is missing.

The number of participating teams is not stated explicitly. From line 232, 232 it seems that it could be deduced to be 34, but it should be stated clearly and from the start.

Understandably individual performance of the teams may remain blinded but a table listing the teams, the participants, and their affiliations would be informative. There are additional names and affiliations provided in the supplementary material under the denomination "Respiratory Viral DREAM Challenge Consortium". Did these individuals also participate in the modeling challenge? It might be possible for the reader to eventually piece together all the information from various parts of the manuscript but he/she should not have to.

Citing the work of the groups who generated the primary data and made them available publicly is good. Listing them in a table by name of the first author, along with their affiliation, size of the datasets contributed etc... would be even better. The amount of effort

required, from obtaining funding each of these clinical studies, having protocols approved, subjects enrolled, samples and clinical data collected, processed etc...., to data deposition is simply enormous.

The number of submissions per sub-challenge could be as high as $34*6=204$. It is therefore not unlikely that significance would be reached by at least one model by chance only. It seems that the number of models submitted may have been accounted for in calculating significance of predictions but this another point that deserves clarification and discussion.

The analysis relied on a disparate collection of datasets, yet no mention is made of batch correction. It may that this was addressed by each team as part of data pre-processing. This should be clarified and/or addressed in the manuscript.

Reviewer #1 (Remarks to the Author):

The paper is well written and an interesting read. I have many difficulties in justifying its publication in this journal. It should be posted back to the challenge pages.

It comes to basically to the same conclusion that the Dream challenge rheumatoid arthritis proposed and published in this journal in 2016. We need to have multiomic datasets of sufficient size to probably achieve any meaningful conclusions.

Thank you for your thoughtful reading of our manuscript, however we respectfully disagree with your assessment. While it is correct that the sample size and study design was insufficient to develop a model which is fully validated for immediate clinical deployment, we have demonstrated an important finding that response to viral exposure can be predicted *even prior to viral exposure*. To the best of our knowledge, this finding is novel, and therefore we think it is of sufficient merit for publication, as it may guide and inspire further research in this field. We have also identified potential biological mechanisms associated with viral susceptibility/resilience. We feel our study serves as important proof-of-principle of general interest to the scientific community and potentially novel findings which we hope will inspire further lines of research by the infectious disease community.

To ensure that the importance of this work is clear to readers, we have highlighted, further, the new findings of our study in the abstract and the main text.

In this particular case of respiratory viruses disease severity, no commonalities were found in the models produced.

We disagree with this assessment. While models had few commonalities with respect to methods used and specific genes selected, they showed significant coherence with respect to biological pathways and processes. This is not unexpected given the high degree of correlation in gene expression data, especially among genes related by pathways or mechanisms, that multiple genes could be selected to represent a specific predictive biological mechanism and minor differences in algorithms or stochasticity (e.g. when performing cross-validations) could result in difference with respect to specific gene predictors, but equivalence with respect to performance and underlying mechanism modeled.

The level of prediction is not clinically useful.

This study focused on identifying biological determinants of susceptibility or resilience to infection that could either be detected at baseline or in the earliest hours after exposure. By its very nature, this is not a clinically useful question. For example, patients who feel well do not typically present to clinical care asking whether they will become sick. Outbreaks pose a notable exception but even that is a clinically rare event compared to the billions of people who develop acute respiratory infections. This study did not intend not to generate a model for clinical deployment therefore, this criticism does not diminish the value of these findings. However, if the purpose of this study were to generate a clinically useful result, the top models

achieved accuracies that are equivalent or better to many tests in clinical use: (AUPR(T_0)=0.958, AUROC(T_0)=0.863).

The problem is complex, RSV is mostly children and Influenza mostly older adults. We should include co-morbidities, age, respiratory capabilities, metabolomic data, among others important modulators of disease severity.

Although we disagree that RSV and influenza have these specific age predilections (both cause widespread infections in both children and adults), we do agree that this study was limited to healthy adults. As such, we are unable to comment on the contribution of these variables. We agree that future studies should include additional information including those suggested. We have included further discussion to this point in the manuscript discussion as follows:

“Additionally, since these studies focused on healthy adults, further data generation should extend to a wider range of subjects with respect to age and health status, as well as tracking and modeling these co-factors.”

The links to the HEME pathway are interesting but remain speculative.

We agree that these observations require future confirmation. We have included a comment to this effect in the discussion:

“This important role of iron in the susceptibility and response to infection may be the mechanism by which HEME METABOLISM genes conferred susceptibility to respiratory viral infection. As such, it represents an important biological pathway potentially offering a means by which an individual’s susceptibility or response to infection can be optimized. Such a relationship should be investigated in future studies of infection susceptibility.”

Reviewer #2 (Remarks to the Author):

This manuscript reports results of a modeling challenge aiming at the identification of blood transcriptional gene signatures predictive of outcomes of exposure to respiratory viruses.

The initiative is excellent and the approach seems well principled. However, some points need to be clarified or addressed.

The overall approach is interesting and appears to have been well thought out but it is also somewhat intricate for outsiders. A figure presenting the overall workflow/process would be helpful.

Thank you for this suggestion. We have added a figure (Figure 1A) to clarify the challenge workflow, which we hope will be helpful to readers.

Some background on how such challenges are prepared and run in general and this one, in particular, would probably be useful as well. It is well explained on the website but could be at least summarized in the manuscript. Some context is missing.

We have also added text to the Methods section of the manuscript describing general framework of the challenge as follows:

“The Challenge was launched and training data were released May 15th, 2016 for participants to use to begin analyzing the data and building their models. In total 38 teams registered for the challenge and 37 participated (Supplementary Table S2). The leaderboards opened approximately 2 months later, and were open for approximately 3 months (July to September) to allow participants to optimize their models with feedback from the scores on the leaderboard data. At the close of this round on September 30th, participating teams were also required to submit code, methodological write-ups, predictor lists, and LOOCVs and doing so qualified participants to be included as authors (either Consortium or by-line) on this manuscript. Participating teams could opt to evaluate their optimized models in the independent test data, which occurred January to February 2017. At the close of the challenge, participating teams were invited to collaborate with the Challenge Organizers to analyze the results. Prior to the launch of the challenge, substantial effort was put forth by the Challenge organizers to collate and vet the data, to determine the feasibility of the Challenge and define the Challenge objectives. For further details on the organizational efforts required to prepare for a challenge, see Saez-Rodriguez et al. (2016) (24).”

The number of participating teams is not stated explicitly. From line 232, 232 it seems that it could be deduced to be 34, but it should be stated clearly and from the start. Understandably individual performance of the teams may remain blinded but a table listing the teams, the participants, and their affiliations would be informative. There are additional names and affiliations provided in the supplementary material under the denomination “Respiratory Viral DREAM Challenge Consortium”. Did these individuals also participate in the modeling challenge? It might be possible for the reader to eventually piece together all the information from various parts of the manuscript but he/she should not have to.

Thank you for your interest in this topic. It is a general policy of DREAM challenges that submissions may be made anonymously, so we do not have complete data on this topic. We have included a table including the list of team names, which phase of the challenge they participated in, and to the degree provided, general affiliations of the team in the supplementary information (Table S2). Participants who have opted to be named are listed either as Consortium authors or named authors depending on whether they also chose to contribute to analyses of the challenge results. We did not feel that it was appropriate to list the names of the individuals on each team, even when available, since the team performances are publicly listed on the Challenge website. This is especially true for individuals who opted not to sign their names to the manuscript. However, we have updated the Author Contribution Statement to clarify which authors were challenge participants. We have also clarified the total number of participating teams in the text.

Citing the work of the groups who generated the primary data and made them available publicly is good. Listing them in a table by name of the first author, along with their affiliation, size of the datasets contributed etc... would be even better. The amount of effort required, from obtaining funding each of these clinical studies, having protocols approved, subjects enrolled, samples and clinical data collected, processed etc...., to data deposition is simply enormous.

We agree that collection of data of this manner is a huge effort and proper attribution of data is important. This was part of the motivation to include these individuals as authors on this manuscript. We have clarified the language in the Author Contribution Statement to make this more clear. Furthermore, we have added a table to summarize the source and attributes of these data in the Supplementary materials (Table S1), which we agree is a useful addition to the manuscript and thank the reviewer for this suggestion.

The number of submissions per sub-challenge could be as high as $34 \times 6 = 204$. It is therefore not unlikely that significance would be reached by at least one model by chance only. It seems that the number of models submitted may have been accounted for in calculating significance of predictions but this is another point that deserves clarification and discussion.

We agree with the importance of accounting for the number of models submitted, however the number quoted above is the maximum number of possible models submitted to the *leaderboard round*, whose purpose was to allow participants to optimize their models prior to final evaluation on the independent test data. Based on feedback from these submissions, each team chose 1 model per timepoint per subchallenge to be evaluated on the independent test data set for a total of 27, 23 and 30 models evaluated in SC2, SC3, and SC1, respectively, as described in the main text of the manuscript. Because these models were assessed independently of the 313 (125, for SC1, 115 for SC2 and 73 for SC3) models submitted to the leaderboard round, it is not necessary to penalize for this number of tests, because under the null hypothesis of no association between the predictions and the true values, the scores from the leaderboard will be independent of the final scores. Further, we have performed no statistical analysis to infer the significance of leaderboard models scores. We have used the leaderboard scores only in a limited manner in this manuscript, namely correlating between (best) leaderboard score and final score (SC1), and correlation between T_0 and T_{24} predictions (cite location). We have clarified the difference between the leaderboard submissions and the final evaluation both in the text, and through Fig 1A, and we hope this will make things more clear to readers.

Returning to the assessment of significance, it is correct that after multiple test corrections, none of the 80 models evaluated on the independent test set achieved individual statistical significance, however the aggregate test of enrichment by subchallenge (SC1, SC2 or SC3), timepoint (T_0 or T_{24}), and metric (AUROC and AUPR for SC1 and SC2, and correlation for SC3) were statistically significant for SC2(T_0 , AUPR) and SC3(T_0 , correlation) with p-value of 0.02 and 0.05 after adjustment for the 10 tests performed (4 each for SC1 and SC2, and 2 for SC3). This analysis shows that while none of the individual scores is significant after correction for multiple testing, the distribution of p-values is stochastically smaller than a uniform(0,1) distribution, as would be expected under the null hypothesis of no association between predictions and true

values, and thus the set of p-values attained by the community are highly unlikely to have been achieved by chance alone.

The analysis relied on a disparate collection of datasets, yet no mention is made of batch correction. It may be that this was addressed by each team as part of data pre-processing. This should be clarified and/or addressed in the manuscript.

As described in the Methods section, challenge participants were given access to both the raw CEL level data, as well as normalized expression in which gene expression arrays were RMA normalized together, adjusting them to the same quantile distribution.

Once teams were provided with the data, they were welcome to correct the expression data in the manner they deemed appropriate. It is of critical importance to note, final evaluation was performed on a held-out study, which was independent of the training set (notably with respect to profiling batch), so any batch-related artifacts present in the training data set would not be predictive in the test set and would have resulted in diminished predictive performance. In fact, we believe this is one of the critical benefits of testing models in independent data sets (cohorts), rather than held-out subsets of the training data studies, because models that overfit artifacts specific to the training data (e.g. batch effects) would be penalized rather than rewarded.

In order to further highlight this information, we have separated this information into its own section entitled "Gene Expression Normalization", so that it is easily found by readers.

REVIEWERS' COMMENTS:

Reviewer #2 (Remarks to the Author):

The response and all changes proposed are satisfactory. There are no further comments or queries from my end.